# Impact of Hypertension on COVID-19 Burden in Kidney Transplant Recipients: An Observational Cohort Study

**DOI:** 10.3390/v14112409

**Published:** 2022-10-30

**Authors:** Isabella Aguiar-Brito, Débora D. de Lucena, Alexandre Veronese-Araújo, Marina P. Cristelli, Hélio Tedesco-Silva, José O. Medina-Pestana, Érika B. Rangel

**Affiliations:** 1Nephrology Division, Department of Medicine, Federal University of São Paulo, Borges Lagoa Street, 591, 6th Floor, Vila Clementino, São Paulo 04038-901, SP, Brazil; 2Hospital do Rim, São Paulo 04038-002, SP, Brazil; 3Hospital Israelita Albert Einstein, São Paulo 05652-900, SP, Brazil

**Keywords:** hypertension, kidney transplant, COVID-19, outcomes

## Abstract

Background: COVID-19 severity is determined by cardiometabolic risk factors, which can be further aggravated by chronic immunosuppression in kidney transplant recipients (KTRs). We aimed to verify the main risk factors related to hypertension (HTN) that contribute to COVID-19 progression and mortality in that population. Methods: Retrospective analysis of 300 KTRs from March 2020 to August 2020 in a single center. We compared the main outcomes between HTN (*n* = 225) and non-HTN (*n* = 75), including admission to the intensive care unit (ICU), development of acute kidney injury (AKI), need for invasive mechanical ventilation or oxygen, and mortality. Results: Of the patients in the study, 57.3% were male, 61.3% were white, the mean age was 52.5 years, and 75% had HTN. Pre-existing HTN was independently associated with higher rates of mortality (32.9%, OR = 1.96, *p* = 0.036), transfer to the ICU (50.7%, OR = 1.94, *p* = 0.017), and AKI with hemodialysis (HD) requirement (40.4%, OR = 2.15, *p* = 0.011). In the hypertensive group, age, diabetes mellitus, heart disease, smoking, glycemic control before admission, C-reactive protein, lactate dehydrogenase, lymphocytes, and D-dimer were significantly associated with COVID-19 progression and mortality. Both lower basal and previous estimated glomerular filtration rates posed KTRs with HTN at greater risk for HD requirement. Conclusions: Therefore, the early identification of factors that predict COVID-19 progression and mortality in KTRs affected by COVID-19 contributes to therapeutic decisions, patient flow management, and allocation of resources.

## 1. Introduction

COVID-19 severity is affected by aging and chronic diseases, such as hypertension (HTN), diabetes mellitus (DM), cardio- and cerebrovascular diseases, obesity, chronic kidney disease (CKD), smoking, neoplasia, and chronic obstructive pulmonary disease (COPD) [1,2,3]. Particularly, the kidney transplant population, not only due to the presence of multiple comorbidities but also due to chronic immunosuppressive regimen, has a higher risk of COVID-19 progression and mortality [4].

HTN is identified as one of the most prevalent comorbidities in patients with COVID-19 [5]. HTN burden increases with age [6] and is frequently associated with underlying comorbidities [7]. Mechanistically, pre-existing cardiovascular diseases may aggravate COVID-19 by the interaction between the viral spike protein of SARS-CoV-2 and angiotensin-converting enzyme 2, which leads to an imbalance of the renin-angiotensin-aldosterone system (RAAS), increased endothelial cell damage, thromboinflammation, and dysregulated immune response [5,8]. 

Given the high prevalence of HTN in the kidney transplant population, we aimed to investigate whether kidney transplant recipients (KTRs) with the diagnosis of HTN had a worse prognosis for COVID-19 when compared to KTRs without HTN in a single center. In addition, we sought to verify the main risk factors associated with COVID-19 outcomes, including mortality, intensive care unit (ICU) admission, need for supplemental oxygen (O_2_), need for invasive mechanical ventilation (IMV), development of acute kidney injury (AKI), and need for hemodialysis (HD).

Early identification of KTRs at high risk of COVID-19 progression provides an effective health system response for disease identification, COVID-19 diagnosis, disease management, monitoring of cases, mortality risk, and resource allocation in future pandemic waves of COVID-19 and other respiratory viruses in general [9,10]. 

## 2. Material and Methods

### 2.1. Study Design and Setting

We conducted a cohort, cross-sectional, observational study at Hospital do Rim, São Paulo, SP, Brazil. We assessed the medical records of patients who were either hospitalized or non-hospitalized with the diagnosis of COVID-19 during the study period of March to August 2020, corresponding to the first wave of COVID-19 in Brazil. We included only patients in whom SARS-CoV-2 was detected by nasopharyngeal swab RT-PCR (reverse transcriptase-polymerase chain reaction). 

The population at risk included 11,875 patients. Of the 590 kidney transplant recipients who were hospitalized, 300 were included in the study. Six were excluded for being a double transplant, four for having lost the graft in the period before COVID-19, four for being a recent transplant and being in delayed graft function at the time of diagnosis of COVID-19, one for not using immunosuppressive drugs due to cancer treatment, one for being underage, and 274 were excluded for missing data due to admission to other services (Appendix A). The Ethics and Research Committee from the Federal University of São Paulo (CAEE35311020.9.0000.8098) approved the study. All the methods were performed following the relevant guidelines and regulations. In addition, this study was performed under the Declaration of Helsinki. Informed consent was obtained from all patients, whereas a waiver was granted for patients who died in other hospitals.

### 2.2. Demographic Data 

We evaluated whether age, sex, race (defined by self-identification), body mass index (BMI), type of donor, time of transplant, as well as the presence of smoking and comorbidities including hypertension, diabetes mellitus (DM), chronic obstructive pulmonary disease (COPD), heart disease, liver disease, and autoimmune disease, were associated with COVID-19-related outcomes. DM was defined by the use of insulin and/or oral antidiabetics, hypertension was defined by whether individuals were taking anti-hypertensive drugs, liver disease by whether hepatitis B or C were diagnosed, and heart disease by whether heart failure and/or coronary artery disease were present. We used the International Classification of Diseases-10.

### 2.3. Laboratory Parameters 

At admission, we evaluated lymphocytes, creatinine, blood glucose, aspartate aminotransferase (AST), alanine aminotransferase (ALT), D-dimer, lactate dehydrogenase (LDH), and C-reactive protein (CRP). We also evaluated the laboratory parameters before admission, including baseline creatinine (mean of the last three measurements), fasting blood glucose (FBG) (last measurement within six months), and glycated hemoglobin (HbA1c) (last measurement within a one year period). The estimated glomerular filtration rate (eGFR) was calculated using the formula defined in the Chronic Kidney Disease Epidemiology Collaboration (CKD-EPI) study: 175 × serum creatinine − 1.154 × age − 0.203 × 1.212 (if black) × 0.742 (if woman) and was expressed in mL/min/1.73 m^2^ of the body surface.

### 2.4. Statistical Analysis

We verified the impact of HTN in kidney transplant recipients with COVID-19 on the outcomes of mortality, transfer to intensive care unit (ICU), acute kidney injury (AKI) classified in accordance with Kidney Disease Improving Global Outcomes (KDIGO) guidelines, need for hemodialysis (HD) and supplemental oxygen (O_2_), and invasive mechanical ventilation (IMV). We then evaluated the impact of hypertension (HTN) in KTRs by dividing them into two groups: HTN (+) and non-HTN or HTN (−).

Independent samples *t*-test and Chi-square test were used to identify the association between HTN and demographic and laboratory parameters, and the outcomes previously mentioned. Data were described as mean ± standard deviation (SD) or median and interquartile range (IQR). Frequencies and percentages were reported for qualitative data. Variables that were univariably associated with a value of *p* ≤ 0.1 were simultaneously included in a multivariable binary logistic regression model to estimate the odds ratios (OR) and 95% confidence interval (CI) between HTN and outcomes. Receiver Operating Characteristic (ROC) curves were used to identify the laboratory parameters associated with COVID-19-related outcomes. To calculate the CRP and LDH cut-off values with better sensitivity and specificity for outcomes, we used the Youden index. We analyzed the data using IBM^®^ SPSS (Statistical Product and Services Solutions, version 18.0, SPSS Inc, Chicago, IL, USA). A *p* value < 0.05 was considered significant for all data analyses.

## 3. Results

HTN was an independent risk factor for mortality (OR = 1.96, 95% CI 1.044–3.682, *p* = 0.036; Appendix A), ICU (OR = 1.94, 95% CI 1.125–3.330, *p* = 0.017; Appendix A), and HD (OR = 2.15, 95% CI 1.188–3.891, *p* = 0.011; Appendix A), but not for oxygen requirement (OR = 0.98, 95% CI 0.581–1.660, *p* = 0.947; Appendix A) and IMV (OR = 1.72, 95% CI 0.958–3.097, *p* = 0.069; Appendix A). Composite analyses including the outcomes of mortality, ICU, and HD also documented HTN as an independent risk factor (OR = 2.18, 95% CI 1.103–4.305, *p* = 0.025; Appendix A). In addition, composite analyses including at least one of these outcomes (mortality or ICU or HD) confirmed HTN as an independent risk factor (OR = 1.99, 95% CI 1.163–3.399, *p* = 0.012; Appendix A). 

When KTRs with HTN were compared to non-HTN, DM (44.4%) and heart disease (13.3%) were found more often (Table 1). To note, only 45 (20%) and 36 (16%) KTR individuals with HTN were taking angiotensin-converting enzyme inhibitors (ACEi) and angiotensin II receptor blockers (ARBs), respectively. 

Despite the presence of HTN, we did not find a difference in laboratory parameters when compared to non-hypertensive KTRs (Table 1). However, clinical outcomes were worse in hypertensive recipients, including higher mortality (OR = 1.96), need for ICU admission (OR = 1.94), development of AKI, particularly stage 3 (OR = 2.20), and need for HD (OR = 2.15) (Table 1). 

Next, we investigated the main risk factors, including demographic data and laboratory parameters that were associated with mortality, transferring to the ICU, and hemodialysis requirement in kidney transplant recipients with HTN, as these outcomes were statistically different when compared to non-HTN individuals. 

In the hypertensive group, age, DM, heart disease, smoking, glycemic control before admission, CRP, LDH, lymphocytes, and D-dimer were significantly associated with mortality (Table 2). In the multivariable analysis, age, heart disease, smoking, previous glycemic control, and lymphopenia remained risk factors for mortality in KTR hypertensive group (Table 2).

On ROC curve analysis, pre-admission glucose values above 116.0 mg/dL resulted in a sensitivity of 64.3% and specificity of 77.1%, with an area under the curve (AUC) of 0.691 (*p* = 0.003) for mortality in hypertensive KTRs, as shown in Figure 1A. Likewise, values greater than 6.3 mg/dL for CRP had a sensitivity of 75.0%, specificity of 56.6%, and AUC of 0.671 (*p* = 0.007) (Figure 1A). LDH had a sensitivity of 60.7%, specificity of 73.5%, and AUC of 0.66 (*p* = 0.011) for values greater than 334 U/L, whereas D-dimer exhibited a sensitivity of 32.1%, specificity of 89.2%, and AUC of 0.574 (*p* = 0.244) for values greater than 2.51 µg/L (Figure 1A). In addition, lymphopenia showed a sensitivity of 71%, specificity of 65.4%, and AUC of 0.675 (*p* < 0.0001) for values less than 673 mm^3^ for mortality in hypertensive patients, as shown in Figure 1B.

When the risk factors for transferring to ICU were analyzed in the hypertensive group, we observed similar risk factors to those found for mortality outcome, including age, heart disease, glycemic control before admission, CRP, LDH, and D-dimer, in addition to eGFR at admission, as shown in Table 3. To note, age, heart disease, glycemic control before admission, LDH, and D-dimer remained risk factors in multivariable analyses, as well as lymphopenia (Table 3). Moreover, the majority of patients transferred to ICU needed IMV (*n* = 83, 72.8%), had AKI (*n* = 100, 87.7%), needed HD (*n* = 85, 74.6%), and died (*n* = 72, 63.2%).

In the analysis of the ROC curve for transferring to ICU, pre-admission glucose and CRP greater than 116.0 mg/dL and 6.1 mg/dL yielded, respectively, sensitivities of 48.1% and 71.2%, specificities of 79.7% and 64.4%, AUC of 0.627 and 0.696 (*p* = 0.021 and *p* < 0.0001), as shown in Figure 2A. LDH and D-dimer yielded, respectively, sensitivities of 53.8% and 32.7%, specificities of 81.4% and 94.9%, AUC of 0.662 and 0.663 (*p* = 0.003, both), for values greater than 334 U/L and 2.46 µg/L (Figure 2A). eGFR at admission and lymphocytes, including values less than 26.5 mL/min/1.73 m² and 686 mm^3^ respectively, yielded sensitivities of 47.1% and 64.4%, specificities of 74.5% and 67.3, and AUC of 0.610 and 0.658 (*p* = 0.007 and *p <* 0.0001) (Figure 2B).

Importantly, renal allograft graft function worsened in the hypertensive group, as 134 (59.6%) individuals had AKI, at different stages according to KDIGO, and 91 (40.4%) needed HD. Therefore, age, deceased donor, liver disease, eGFR at baseline and admission, glycemic control before admission, CRP, and LDH were associated with the need for HD (shown in Table 4). Nevertheless, only age was statistically significant in the multivariable analysis, as shown in Table 4. Among the hypertensive patients who required HD, 85 (93.4%) were transferred to the ICU, 77 (84.6%) needed IMV, and 66 (72.5%) died. 

When HD was required, for glycemia control before admission and CRP, values greater than 98.5 and 4.56 mg/dL, the analyses of the ROC curve yielded sensitivities of 66.0% and 74.0%, specificities of 60.7% and 63.1%, and AUC of 0.638 and 0.678 (*p* = 0.008 and *p* = 0.001, respectively), while for LDH, sensitivity was 58.0% and specificity was 73.8% for values above 334 mg/dL (*p* = 0.002) and AUC was 0.662 as shown in Figure 3A. In addition, eGFR at baseline and admission showed, respectively, sensitivities of 39.6% and 49.5%, specificities of 87.3% and 81.3%, and AUC of 0.621 and 0.678 (*p* = 0.002 and *p* < 0.0001) for values less than 29.5 and 24 mL/min/1.73 m² for the need for HD (Figure 3B).

We next evaluated the outcomes in the non-hypertensive KTR group (*n* = 75) by using univariable analyses. Thus, age (*p* = 0.006), eGFR at admission (*p* = 0.023), CRP (*p* = 0.021), and LDH (*p* = 0.019) were also associated with the mortality outcome, as shown in Appendix A. In addition, DM (42.3% vs. 12.2%, *p* = 0.005), LDH (*p* = 0.004), and lymphopenia (*p* = 0.017) were associated with admission to ICU of the non-hypertensive KTRs, as documented in Appendix A. For HD requirement, eGFR at admission (23 vs. 43 mL/min/1.73 m^2^, *p* = 0.001) and LDH (*p* = 0.009) were risk factors, as shown in Appendix A. Therefore, after performing multivariable analyses in the non-hypertensive KTR group, age was associated with mortality (Appendix A), as verified in the hypertensive KTR group (Table 2). LDH, a marker of tissue damage, was a risk factor in both non-HTN and HTN patients for mortality (Table 2 and Appendix A), need for ICU transferring (Table 3 and Appendix A), and HD requirement (Table 4 and Appendix A). To note, in non-hypertensive KTRs, DM was an important risk factor for stratifying these patients for ICU requirement (Appendix A). Low rates of eGFR at admission were a risk factor for HD requirement in both non-HTN (Appendix A) and HTN patients (Table 4).

## 4. Discussion

Our study showed that KTRs with HTN had higher rates of mortality, need for ICU admission, development of AKI, and need for HD when affected by COVID-19. Even though the need for IMV was not different between hypertensive and non-hypertensive recipients, the former required IMV more often. The risk factors that were associated with those worse outcomes in KTRs with HTN comprised demographic and laboratory data, including age, the presence of other comorbidities, particularly DM and heart disease, higher levels of previous glucose, CRP, LDH, and D-dimer, and lower levels of eGFR and lymphocytes on admission. 

As previously documented in the literature throughout the first year of the pandemic, KTRs presented high rates of disease progression and mortality [11], even when compared to non-transplant patients [12]. However, the association between solid organ transplant (SOT) and increased mortality was debated in the literature, as the presence of comorbidities per se can explain the higher rates of mortality. Therefore, with the application of a propensity score, two distinct studies showed that mortality of KTRs and the general population were similar, suggesting that the worse outcomes of KTRs were mainly explained by the number of comorbidities and not entirely attributable to chronic immunosuppression [13,14]. A recent systematic review and meta-analysis supported these findings [15]. 

In line with these findings, the National COVID Cohort Collaborative documented that SOT recipients with comorbidities, in particular, HTN, DM, coronary artery disease, and CKD, were tested more often for COVID-19 when compared to patients without these comorbidities [16]. In addition, COVID-19 was associated with AKI, graft rejection, and graft failure, as well as with the occurrence of major adverse cardiac events (acute myocardial infarction, angina, stent occlusion/thrombosis, stroke, transient ischemic attack, congestive heart failure, or death) in SOT setting [16]. 

When we compared our data to an analysis of 2,054 Brazilian non-transplant individuals during the same period of the COVID-19 pandemic, we verified that our population was younger (53 years old vs. 58 years old) and presented more comorbidities (HTN: 74.7% vs. 52.9%; DM: 39% vs. 29.2%; obesity: 21.7% vs. 17.2%) [17]. Despite these demographic differences, we observed similar rates of ICU transferring (46.7% vs. 47.6%) and IMV (34% vs. 32.5%), yet we encountered higher in-hospital mortality rates (29.7% vs. 22%) and need for HD (36.3% vs. 12.2%) [17]. These findings may be explained by the greater number of comorbidities and lower kidney function in the renal transplant population. Likewise, a Brazilian longitudinal cohort study of 398,063 admissions in public hospitals of non-transplanted individuals with COVID-19 of all ages documented a case-fatality rate of 21.7%, decreasing from 31.8% to 18.2% following the same time of our study (March to October 2020) [18]. 

HTN is an independent risk factor for COVID-19 progression and mortality [19], as we have also demonstrated. Conversely, the British and Irish Hypertension Society documented, after examining over 17 million health records in England, that HTN or a recorded blood pressure ≥ 140/90 mmHg were not associated with higher mortality rates after full adjustment [20]. Therefore, the isolated diagnosis of HTN was associated with a slightly higher risk of mortality (HR = 1.07), which is explained by the strong age-related association. 

Age is an independent risk factor for COVID-19 mortality [21,22], which may be explained by age-related immunosenescence [23]. Moreover, people over 60 years old have a significantly higher prevalence of cardio-metabolic risk factors, such as HTN, DM, obesity, and dyslipidemia, which also contribute to the worsening of COVID-19 [24]. In a multicenter study with 1,303 hospitalized patients, 39.9% had a cardio-metabolic disease and, when compared to patients without the cardio-metabolic disease, those patients had more COVID-19-related complications, including ARDS, AKI, secondary infection, hypoproteinemia, and coagulopathy [25]. In addition, the cardio-metabolic group had higher incidences of COVID-19 progression, including admission to ICU, IMV, and mortality. Similarly, in a larger cohort with 29,995 participants, cardio-metabolic risk factors, adjusted for age and gender, were still associated with COVID-19 hospitalization and mortality, comprising obesity (aOR = 4.09), pre-diabetes (aOR = 2.56), DM (aOR = 3.96), sedentary time (aOR per hour/day increase = 1.10) and grade 2 HTN (aOR = 2.44), and high-density lipoprotein cholesterol (aOR per mmol/L increase = 0.33) [26]. These findings highlight the paramount importance of adherence to pharmacological and non-pharmacological approaches to decrease the cardio-metabolic syndrome burden, and ultimately, the COVID-19 severity, in particular, in the kidney transplant population. 

Likewise, the presence of comorbidities may also adversely impact the COVID-19 severity in the transplant setting. Thus, in our group of KTRs with HTN, 44.4% also had DM. A combined prevalence of HTN and DM appears to confer the greatest mortality (OR = 2.75) when HTN is analyzed separately (OR = 1.69) [27]. On top of these findings, hyperglycemia has emerged as an important risk factor for COVID-19 progression. In a study of 3,854 patients, glycemic levels greater than >170 mg/dL were also associated with worse outcomes in COVID-19, including an increase in intubation rate (OR = 15.6) and mortality (OR = 3.6), longer hospitalization time, and increased risk for developing ARDS (OR = 9.3) [28]. Using propensity-score matching, our group also documented that higher previous fasting blood glucose was associated with worse outcomes on KTRs independently of DM status [29]. As SARS-CoV-2 induces adipose tissue dysfunction by binding to ACE2 receptor [8], the reduction of adiponectin and adiponectin/leptin ratio is associated with the increase in type I interferon signaling pathway and activation of the innate immune response, which leads to insulin resistance and, consequently, hyperglycemia [28]. These findings may also explain the association between previous glucose control and COVID-19 adverse outcomes in our KTRs with HTN, in particular, ICU transferring, need for HD, and mortality. To note, in KTRs without HTN, we did not verify this association. 

From the pathophysiology perspective of COVID-19, SARS-CoV-2 induces direct cell toxicity, dysregulation of RAAS and kallikrein-kinin system, endothelial cell damage associated with thromboinflammation and thromboembolic events, dysregulation of the immune system characterized by hyperactivation of the innate immune system, hyper inflammation caused by inhibition of interferon signaling, T cell lymphodepletion by exhaustion, and the production of proinflammatory cytokines, particularly IL-6 and TNFα, which ultimately lead to cytokine storm [30,31]. 

Therefore, ACE2 triggers the entry of SARS-CoV-2 into host cells [8] and is highly expressed in the heart, kidneys, and lungs, which explains SARS-CoV-2 tropism and damage to these organs [32]. Strikingly, ACE2 is over-expressed within the lungs of individuals with comorbidities [33], contributing to COVID-19 progression and mortality, as observed in our population of KTRs with HTN, DM, CKD, and cardiac disease. Therefore, viral pathogen-associated molecular patterns (PAMPs) bind to Toll-like receptors-4 located in alveolar macrophage and activate adaptor protein MyD88-TNF receptor-associated factor (TRAF) 6 pathway and nuclear factor kappa B (NF-κB) translocation into the nucleus, as well as the inflammasome, and ultimately the proinflammatory cytokine release, as reviewed elsewhere [34]. Smoking can also augment the inflammatory microenvironment in the lungs and contributes to COVID-19 mortality [35], as we observed in our study. 

In addition, the presence of those comorbidities is associated with chronic endothelial dysfunction, which is aggravated by SARS-CoV-2 infection and endotheliitis [8]. Endothelial dysfunction is related to vascular and cardiac complications in patients with COVID-19 as these patients exhibited substantially lower flow-mediated dilation of peripheral arteries during the post-infection stage [36]. Therefore, temporal changes in the endothelium-mediated dilation of peripheral arteries, found in acute COVID-19 to the post-infection stage, suggest that endothelial vascular dysfunction may be a chronic complication of this disease [37].

In clinical practice, the interaction of SARS-CoV-2 with the ACE2 receptor did not affect the decision of using RAAS blockers, primarily ACEi and ARBs. These drugs neither increase ACE2 expression within the motile cilia of upper and lower airway epithelial cells [38] nor increase the risk of COVID-19 and the likelihood of a positive test [39,40], even when adjusted for age, gender, and cardiovascular risk [41]. Moreover, poor blood-pressure control was associated with worse outcomes in HTN patients with COVID-19, indicating that these patients may have more advanced atherosclerosis and target organ damage [42]. Therefore, anti-hypertensive drugs should not be discontinued in hypertensive patients during the COVID-19 pandemic.

In addition, early identification of laboratory parameters associated with SARS-CoV-2-mediated thromboinflammation is a key aspect to manage KTRs during the pandemic. Longitudinal analyses of these markers indicate different degrees of inflammation (increased levels of IL-6 and CRP), infection (increased levels of ferritin and neutrophils and decreased levels of lymphocytes), thrombosis (increased D-dimer and thrombin associated with decreased fibrinolysis), and tissue damage (increased LDH, AST, and ALT, and decreased fibrinogen) [30]. 

Importantly, as we verified in our group of KTRs with HTN who died, increased levels of CRP [43], LDH [44], and D-dimer can be used as prognostic factors in COVID-19. These markers, as well as procalcitonin and high sensitivity-troponin I, are also useful tools for risk-stratifying KTRs with COVID-19 [11,22]. In addition, lymphopenia is a surrogate marker of poor prognosis in COVID-19 [11,45], which prompts the reduction or withdrawal of immunosuppressive drugs used in transplant settings, particularly antimetabolites [11]. 

AKI is paramount in determining the severity of COVID-19 and is independently associated with progression and mortality, including all stages and the need for HD [46]. Predictors of AKI include age, black race, male gender, HTN, DM, obesity, and lower eGFR [47]. In our population, 58% of our population developed AKI regardless of the presence of HTN, although in KTRs with HTN the degree of AKI was more severe. Similar rates of AKI were found in KTRs during the COVID-19 pandemic [11,48,49]. Systemic inflammation and kidney function, which is usually lower in deceased donors, and hepatic diseases, pose these patients at higher risk of AKI [22,49], as we also verified in our population.

Our study has some limitations, as we performed a retrospective analysis in a single center, the number of patients that were excluded due to lack of data, other laboratory parameters were not available (such as immune profiling, cardiac and hematologic markers), the low number of patients using ACE inhibitors or ARBs, and the lack of use of drugs that were later demonstrated to have efficacy during the pandemic, such as dexamethasone and tocilizumab.

## 5. Conclusions

In conclusion, demographic and laboratory parameters may be used to improve risk stratification at hospital admission in KTRs with HTN. Therefore, the early identification of factors that predict COVID-19 progression and mortality in KTRs affected by COVID-19 contributes to therapeutic decisions, patient flow management, and allocation of resources.

## Figures and Tables

**Figure 1 viruses-14-02409-f001:**
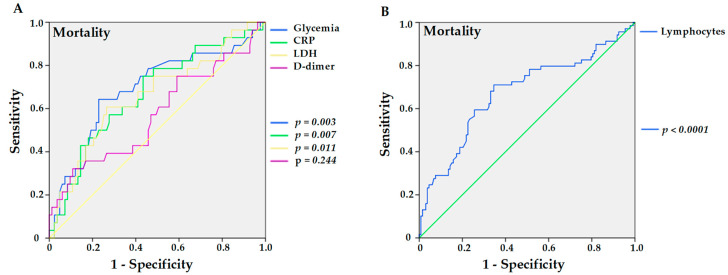
Sensitivities and specificities of higher levels of previous glycemia (*p* = 0.003), C-reactive protein (CRP, *p* = 0.007), lactate dehydrogenase (LDH; *p* = 0.011), and D-dimer (*p* = 0.244) (**A**), and lower levels of lymphocytes (*p* < 0.0001) (**B**) in hypertensive kidney transplant recipients with COVID-19 for the outcome of mortality.

**Figure 2 viruses-14-02409-f002:**
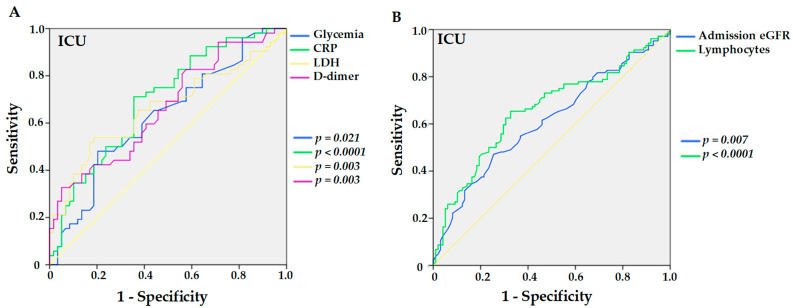
Sensitivities and specificities of higher levels of previous glycemia (*p* = 0.021), C-reactive protein (CRP, *p* < 0.0001), lactate dehydrogenase (LDH, *p* = 0.003), and D-dimer (*p* = 0.003) (**A**), and lower levels of estimated glomerular ratio (eGFR) at admission (*p* = 0.007) and lymphocytes (*p* < 0.0001) (**B**) in hypertensive kidney transplant recipients with COVID-19 for the outcome of transferring to the ICU.

**Figure 3 viruses-14-02409-f003:**
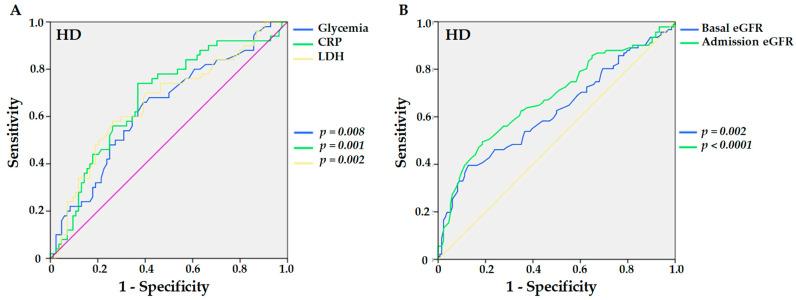
Sensitivities and specificities of higher levels of previous glycemia (*p* = 0.008), C-reactive protein (CRP, *p* = 0.001), and lactate dehydrogenase (LDH, *p* = 0.002) (**A**), and lower levels of eGFR at baseline (*p* = 0.002) and at admission (*p* < 0.0001) (**B**) in hypertensive kidney transplant recipients with COVID-19 for the outcome of the need for hemodialysis (HD).

**Table 1 viruses-14-02409-t001:** Demographic data, laboratory data, and outcomes in kidney transplant recipients according to the presence of hypertension (HTN).

	HTN (+) (*n* = 225, 75%)	HTN (−) (*n* = 75, 25%)	Total (*N* = 300, 100%)	Univariable Analysis	Multivariable Analysis
**Demographic data**					
Age (years)	53.2 ± 11.4	50.4 ± 14.1	52.5 ± 12.2	1.02 (0.997–1.042, *p* = 0.085)	1.00 (0.979–1.026, *p* = 0.862)
Male (*n*, %)	136 (60.4)	36 (48.0)	172 (57.3)	1.66 (0.978–2.801, *p* = 0.060)	1.76 (1.018–3.041, ***p* = 0.043**)
Race (*n*, %)				0.74 (0.425–1.275, *p* = 0.274)	
White	134 (59.6)	50 (66.7)	184 (61.3)		
Black/brown	85 (37.8)	25 (33.3)	110 (36.7)		
Transplant time (months)	68 (33.0;142.0)	101 (39.5;142.5)	81.5 (33.8;142.3)	1.00 (0.995–1.002, *p* = 0.436)	
Donor type (*n*, %)				1.62 (0.927–2.830, *p* = 0.090)	1.47 (0.820–2.635, *p* = 0.196)
Living	58 (25.8)	27 (36.0)	85 (28.3)		
Deceased	167 (74.2)	48 (64.0)	215 (71.7)		
BMI (kg/m²)	27.2 ± 4.7	26.1 ± 5.3	26.9 ± 4.9	1.05 (0.991–1.113, *p* = 0.100)	
BMI ≥ 25 (*n*, %)	93 (41.3)	27 (36.0)	120 (40.0)	1.19 (0.682–2.063, *p* = 0.546)	
BMI ≥ 30 (*n*, %)	52 (23.1)	13 (17.3)	65 (21.7)	1.40 (0.710–2.757, *p* = 0.332)	
DM (*n*, %)	100 (44.4)	17 (22.7)	117 (39.0)	2.73 (1.496–4.979, ***p* = 0.001**)	2.38 (1.250–4.510, ***p* = 0.008**)
COPD (*n*, %)	7 (3.1)	2 (2.7)	9 (3.0)	1.17 (0.238–5.768, *p* = 0.845)	
Heart disease (*n*, %)	30 (13.3)	2 (2.7)	32 (10.7)	5.62 (1.309–24.903, ***p* = 0.020**)	4.24 (0.952–18.897; *p* = 0.058)
Neoplasia (*n*, %)	15 (6.7)	6 (8.0)	21 (7.0)	0.82 (0.307–2.200, *p* = 0.696)	
Liver disease (*n*, %)	9 (4.0)	0 (0)	9 (3.0)	- (-, *p* = 0.999)	
Autoimmune disease (*n*, %)	4 (1.8)	2 (2.7)	6 (2.0)	0.66 (0.119–3.681, *p* = 0.636)	
Smoking (*n*, %)	48 (21.3)	14 (18.7)	62 (20.7)	1.24 (0.626–2.444, *p* = 0.540)	
**Laboratory data**					
Basal eGFR	47 (30;64)	48 (32;65)	46.8 (30.8;64.2)	1.00 (0.990–1.011, *p* = 0.936)	
Admission eGFR	34 (19;49)	35 (22;52)	34 (20;50)	1.00 (0.985–1.009, *p* = 0.633)	
Previous glycemia (mg/dL)	97 (86.3;136)	94 (82;108)	96 (84;121)	1.00 (0.999–1.009, *p* = 0.158)	
CRP (mg/dL)	6.3 (2.1;13.2)	5.2 (1.2;10.6)	5.7 (2.0;12.6)	1.03 (0.988–1.063, *p* = 0.187)	
LDH (U/L)	288.5 (227;405.5)	273 (191;354)	287 (220;395)	1.00 (1.000–1.003, *p* = 0.103)	
Lymphocytes (mm³)	695(461.8;1195.3)	852 (520;1122)	738 (468;1174.5)	1.00 (1.000–1.000, *p* = 0.795)	
D-dimer (μg/L)	1.2 (0.6;2.3)	1.2 (0.5;1.8)	1.2 (0.6;2.3)	1.11 (0.987–1.258, *p* = 0.080)	
AST (U/L)	28 (20;40)	32 (24;43)	28 (21;41)	1.00 (0.991–1.006, *p* = 0.663)	
ALT (U/L)	21 (14;32.5)	20 (16;28)	21 (15;32)	1.00 (0.990–1.014, *p* = 0.768)	
**Outcomes**					
Mortality (*n*, %)	74 (32.9)	15 (20.0)	89 (29.7)	1.96 (1.044–3.682, ***p* = 0.036**)	1.67 (0.572–4.850, *p* = 0.349)
ICU (*n*, %)	114 (50.7)	26 (34.7)	140 (46.7)	1.94 (1.125–3.330, ***p* = 0.017**)	1.83 (0.724–4.646, *p* = 0.201)
O_2_ (*n*, %)	122 (54.2)	41 (54.7)	163 (54.3)	0.98 (0.581–1.660, *p* = 0.947)	
IMV (*n*, %)	83 (36.9)	19 (25.3)	102 (34.0)	1.72 (0.958–3.097, *p* = 0.069)	0.47 (0.124–1.781, *p* = 0.267)
AKI (*n*, %)	134 (59.6)	40 (53.3)	174 (58.0)	1.29 (0.761–2.180, *p* = 0.345)	
Stage 1	25 (11.1)	13 (17.3)	38 (12.7)	0.60 (0.288–1.235, *p* = 0.164)	
Stage 2	9 (4.0)	7 (9.3)	16 (5.3)	0.41 (0.145–1.128, *p* = 0.084)	0.44 (0.146–1.340; *p* = 0.149)
Stage 3	100 (44.4)	20 (26.7)	120 (40.0)	2.20 (1.237–3.911, ***p* = 0.007**)	1.54 (0.383–6.211; *p* = 0.542)
HD (*n*, %)	91 (40.4)	18 (24.0)	109 (36.3)	2.15 (1.188–3.891, ***p* = 0.011**)	1.12 (0.221–5.637, *p* = 0.895)

HTN: hypertension; BMI: body mass index; DM: diabetes mellitus; COPD: chronic obstructive pulmonary disease; eGFR: estimated glomerular filtration rate (in mL/min/1.73 m^2^); CRP: C-reactive protein; LDH: lactate dehydrogenase; AST: aspartate aminotransferase; ALT: alanine aminotransferase; ICU: intensive care unit; O_2_: use of supplemental oxygen; IVM: invasive mechanical ventilation; AKI: acute kidney injury; HD: hemodialysis. All variables are median and IQR, except age and BMI, which are means ± SD. Univariable and multivariable analyses are represented by odds ratios, 95% confidence intervals, and *p*-values, respectively.

**Table 2 viruses-14-02409-t002:** Risk factors for mortality in kidney transplant recipients with hypertension (HTN).

	Not Alive (*n* = 74, 32.9%)	Alive (*n* = 151, 67.1%)	Univariable Analysis	Multivariable Analysis
**Demographic Data**				
Age (years)	58.6 ± 10.5	50.5 ± 11.0	1.07 (1.041–1.104, ***p* < 0.0001**)	1.06 (1.026–1.100, ***p* = 0.001**)
Male (*n*, %)	44 (59.5)	92 (60.9)	0.94 (0.533–1.659, *p* = 0.832)	
Race (*n*, %)			1.18 (0.665–2.081, *p* = 0.577)	
White	46 (62.2)	88 (58.3)		
Black/brown	24 (32.4)	61 (40.4)		
Transplant time (months)	78 (41;125.8)	65 (30;143.5)	1.00 (0.997–1.005, *p* = 0.654)	
Donor type (*n*, %)			1.76 (0.893–3.477, *p* = 0.102)	
Living	14 (18.9)	44 (29.1)		
Deceased	60 (81.1)	107 (70.9)		
BMI (kg/m²)	27.2 ± 4.4	27.1 ± 4.9	1.00 (0.945–1.067, *p* = 0.894)	
BMI ≥ 25 (*n*, %)	32 (43.2)	61 (40.4)	1.21 (0.677–2.144, *p* = 0.526)	
BMI ≥ 30 (*n*, %)	17 (23.0)	35 (23.2)	1.04 (0.532–2.019, *p* = 0.915)	
DM (*n*, %)	40 (54.1)	60 (39.7)	1.78 (1.018–3.128, ***p* = 0.043**)	0.96 (0.461–1.994, *p* = 0.911)
COPD (*n*, %)	2 (2.7)	5 (3.3)	0.81 (0.154–4.283, *p* = 0.805)	
Heart disease (*n*, %)	17 (23.0)	13 (8.6)	3.17 (1.444–6.943, ***p* = 0.004**)	3.70 (1.302–10.526, ***p* = 0.014**)
Neoplasia (*n*, %)	7 (9.5)	8 (5.3)	1.87 (0.650–5.364, *p* = 0.246)	
Liver disease (*n*, %)	5 (6.8)	4 (2.6)	2.66 (0.693–10.227, *p* = 0.154)	
Autoimmune disease (*n*, %)	1 (1.4)	3 (2.0)	0.68 (0.069–6.610, *p* = 0.736)	
Smoking (*n*, %)	22 (29.7)	26 (17.2)	2.12 (1.071–4.180, ***p* = 0.031**)	2.20 (1.051–4.616, ***p* = 0.036**)
**Laboratory Data**				
Basal eGFR	49 (25;63)	46 (32;63)	1.00 (0.988–1.010, *p* = 0.870)	
Admission eGFR	30 (17;46)	35 (22;50)	1.00 (0.982–1.008, *p* = 0.434)	
Previous glycemia (mg/dL)	115 (93;191)	92 (83;114)	1.01 (1.006–1.017, ***p* < 0.0001**)	1.02 (1.007–1.029, ***p* = 0.002**)
CRP (mg/dL)	13.2 (5.5;19,0)	4.4 (1.7;10.3)	1.08 (1.038–1.119, ***p* < 0.0001**)	1.03 (0.964–1.095, *p* = 0.410)
LDH (U/L)	360 (271.5;474.5)	270 (224;356)	1.00 (1.000–1.003, ***p* = 0.049**)	1.00 (0.999–1.004, *p* = 0.193)
Lymphocytes (mm³)	524 (348;789)	819 (570;1230)	1.00 (0.999–1.000, ***p* = 0.039**)	1.00 (0.997–1.000, ***p* = 0.043**)
D-dimer (μg/L)	1.5 (0.7;2.8)	1.1 (0.6;2.1)	1.10 (1.020–1.194, ***p* = 0.014**)	1.19 (0.988–1.426, *p* = 0.067)
AST (U/L)	29 (20;41.5)	27 (21;39)	1.00 (0.992–1.009, *p* = 0.889)	
ALT (U/L)	22.5 (13.8;32.3)	21 (15;33)	1.00 (0.990–1.013, *p* = 0.836)	

HTN: hypertension; BMI: body mass index; COPD: chronic obstructive pulmonary disease; eGFR: estimated glomerular filtration rate (in mL/min/1.73 m^2^); CRP: C-reactive protein; LDH: lactate dehydrogenase; AST: aspartate aminotransferase; ALT: alanine aminotransferase. Age and BMI are means ± SD, while transplant time, basal eGFR, admission eGFR, previous glycemia, CRP, LDH, lymphocytes, D-dimer, AST, and ALT are medians and IQR. Univariable and multivariable analyses are indicated as odds ratios, 95% confidence intervals, and *p*-values.

**Table 3 viruses-14-02409-t003:** Risk factors for intensive care unit (ICU) admission in kidney transplant recipients with hypertension (HTN).

	ICU (*n* = 114, 50.7%)	No ICU (*n* = 111, 49.3%)	Univariable Analysis	Multivariable Analysis
**Demographic Data**				
Age (years)	56.9 ± 11.1	49.3 ± 10.5	1.07 (1.039–1.096, ***p* < 0.0001**)	1.05 (1.018–1.081, ***p* = 0.002**)
Male (*n*, %)	65 (57.0)	71 (64.0)	0.75 (0.437–1.278, *p* = 0.287)	
Race (*n*, %)			1.17 (0.686–1.991, *p* = 0.567)	
White	70 (61.4)	64 (57.7)		
Black/brown	39 (34.2)	46 (41.4)		
Transplant time (months)	84 (37.5;137)	61 (29;142.5)	1.00 (0.997–1.005, *p* = 0.549)	
Donor type (*n*, %)			1.04 (0.570–1.884, *p* = 0.906)	
Living	29 (25.4)	29 (26.1)		
Deceased	85 (74.6)	82 (73.9)		
BMI (kg/m²)	27.4 ± 4.6	26.9 ± 4.8	1.02 (0.967–1.084, *p* = 0.422)	
BMI ≥ 25 (*n*, %)	48 (42.1)	45 (40.5)	1.07 (0.622–1.830, *p* = 0.815)	
BMI ≥ 30 (*n*, %)	29 (25.4)	23 (20.7)	1.31 (0.698–2.450, *p* = 0.401)	
DM (*n*, %)	53 (46.5)	47 (42.3)	1.18 (0.699–2.003, *p* = 0.531)	
COPD (*n*, %)	5 (4.4)	2 (1.8)	2.50 (0.475–13.164, *p* = 0.280)	
Heart disease (*n*, %)	23 (20.2)	7 (6.3)	3.76 (1.540–9.159, ***p* = 0.004**)	4.84 (1.316–17.829, ***p* = 0.018**)
Neoplasia (*n*, %)	11 (9.6)	4 (3.6)	2.86 (0.881–9.259, *p* = 0.080)	4.99 (0.547–45.405, *p* = 0.154)
Liver disease (*n*, %)	6 (5.3)	3 (2.7)	2.00 (0.488–8.203, *p* = 0.336)	
Autoimmune disease (*n*, %)	3 (2.6)	1 (0.9)	2.97 (0.305–29.022, *p* = 0.349)	
Smoking (*n*, %)	31 (27.2)	17 (15.3)	1.97 (0.994–3.890, *p* = 0.052)	2.10 (1.010–4.360, ***p* = 0.047**)
**Laboratory Data**				
Basal eGFR	44 (27;60)	48 (33;64.7)	1.00 (0.985–1.005, *p* = 0.327)	
Admission eGFR	29 (17;45)	39 (26;52)	0.99 (0.973–0.997, ***p* = 0.018**)	0.98 (0.956–1.004, *p* = 0.094)
Previous glycemia (mg/dL)	108.5 (89;163.8)	92 (83;107.5)	1.01 (1.006–1.018, ***p* < 0.0001**)	1.01 (1.001–1.022, ***p* = 0.026**)
CRP (mg/dL)	10.1 (3.7;17.3)	3.7 (1.4;9.6)	1.08 (1.038–1.124, ***p* < 0.0001**)	1.01 (0.947–1.079, *p* = 0.739)
LDH (U/L)	351.5 (241;483.5)	250.5 (219.8;317.8)	1.00 (1.002–1.007, ***p* = 0.001**)	1.00 (1.001–1.008, ***p* = 0.024**)
Lymphocytes (mm³)	585 (373.8;903.8)	861 (578.5;1289.8)	1.00 (0.999–1.000, *p* = 0.095)	1.00 (0.998–1.000, ***p* = 0.025**)
D-dimer (μg/L)	1.6 (0.8;2.9)	0.9 (0.5;1.7)	1.17 (1.041–1.303, ***p* = 0.008**)	1.70 (1.102–2.626, ***p* = 0.016**)
AST (U/L)	29 (20;42)	27 (21;37)	1.01 (0.997–1.020, *p* = 0.146)	
ALT (U/L)	22 (14;31)	21 (15;33.5)	1.00 (0.990–1.013, *p* = 0.798)	

HTN: hypertension; BMI: body mass index; COPD: chronic obstructive pulmonary disease; eGFR: estimated glomerular filtration rate (in mL/min/1.73 m^2^); CRP: C-reactive protein; LDH: lactate dehydrogenase; AST: aspartate aminotransferase; ALT: alanine aminotransferase. Age and BMI are means ± SD, while transplant time, basal eGFR, admission eGFR, previous glycemia, CRP, LDH, lymphocytes, D-dimer, AST, and ALT are medians and IQR. Univariable and multivariable analyses are indicated as odds ratios, 95% confidence intervals, and *p*-values.

**Table 4 viruses-14-02409-t004:** Risk factors for the need for hemodialysis (HD) in kidney transplant recipients with hypertension (HTN).

	HD (*n* = 91, 40.4%)	No HD (*n* = 134, 59.6%)	Univariable Analysis	Multivariable Analysis
**Demographic Data**				
Age (years)	56.4 ± 11.0	50.9 ± 11.3	1.05 (1.019–1.072, ***p* = 0.001**)	1.05 (1.015–1.077, ***p* = 0.003**)
Male (*n*, %)	56 (61.5)	80 (59.7)	1.08 (0.626–1.863, *p* = 0.782)	
Race (*n*, %)			1.24 (0.719–2.142, *p* = 0.438)	
White	57 (62.6)	77 (57.5)		
Black/brown	30 (33.0)	55 (41.0)		
Transplant time (months)	85 (41;130)	62 (29;143.8)	1.00 (0.997–1.004, *p* = 0.683)	
Donor type (*n*, %)			1.92 (1.009–3.649, ***p* = 0.047**)	1.15 (0.503–2.643, *p* = 0.737)
Living	17 (18.7)	41 (30.6)		
Deceased	74 (81.3)	93 (69.4)		
BMI (kg/m²)	27.3 ± 4.5	27.1 ± 4.9	1.01 (0.950–1.067, *p* = 0.815)	
BMI ≥ 25 (*n*, %)	37 (40.7)	56 (41.8)	0.95 (0.549–1.649, *p* = 0.859)	
BMI ≥ 30 (*n*, %)	23 (25.3)	29 (21.6)	1.23 (0.653–2.306, *p* = 0.526)	
DM (*n*, %)	45 (49.5)	55 (41.0)	1.41 (0.822–2.402, *p* = 0.214)	
COPD (*n*, %)	4 (4.4)	3 (2.2)	2.01 (0.439–9.191, *p* = 0.369)	
Heart disease (*n*, %)	17 (18.7)	13 (9.7)	2.14 (0.982–4.655, *p* = 0.056)	2.11 (0.772–5.756, *p* = 0.146)
Neoplasia (*n*, %)	8 (8.8)	7 (5.2)	1.75 (0.611–5.004, *p* = 0.298)	
Liver disease (*n*, %)	7 (7.7)	2 (1.5)	5.50 (1.116–27.109, ***p* = 0.036**)	2.41 (0.445–13.048, *p* = 0.307)
Autoimmune disease (*n*, %)	2 (2.2)	3 (2.2)	1.48 (0.205–10.724, *p* = 0.696)	
Smoking (*n*, %)	26 (28.6)	22 (16.4)	1.78 (0.917–3.470, *p* = 0.088)	1.69 (0.835–3.407, *p* = 0.145)
**Laboratory Data**				
Basal eGFR	41 (21;58)	50 (35;68)	0.99 (0.974–0.996, ***p* = 0.009**)	1.00 (0.977–1.017, *p* = 0.753)
Admission eGFR	25 (13;43)	40 (26;53)	0.97 (0.960–0.988, ***p* < 0.0001**)	0.98 (0.951–1.002, *p* = 0.075)
Previous glycemia (mg/dL)	108.5 (90;164.5)	92.5 (83;112.8)	1.01 (1.005–1.015, ***p* < 0.0001**)	1.01 (0.999–1.017, *p* = 0.075)
CRP (mg/dL)	10.1 (4.5;17.1)	3.9 (1.7;10.5)	1.05 (1.016–1.088, ***p* = 0.004**)	1.03 (0.975–1.084, *p* = 0.306)
LDH (U/L)	357.5 (247.3;488.3)	268 (221;347)	1.00 (1.000–1.004, ***p* = 0.041**)	1.00 (0.999–1.003, *p* = 0.498)
Lymphocytes (mm³)	534.5 (371;834)	847 (584.3;1260.8)	1.00 (0.999–1.000, *p* = 0.161)	
D-dimer (μg/L)	1.5 (0.7;2.7)	1.1 (0.6;2.1)	1.07 (0.990–1.153, *p* = 0.087)	1.17 (0.986–1.385, *p* = 0.073)
AST (U/L)	29 (20;42)	27 (21;38)	1.00 (0.993–1.009, *p* = 0.880)	
ALT (U/L)	23 (14;32)	21 (15;32.8)	1.00 (0.988–1.011, *p* = 0.941)	

HTN: hypertension; BMI: body mass index; COPD: chronic obstructive pulmonary disease; eGFR: estimated glomerular filtration rate (in mL/min/1.73 m^2^); CRP: C-reactive protein; LDH: lactate dehydrogenase; AST: aspartate aminotransferase; ALT: alanine aminotransferase. Age and BMI are means ± SD, while transplant time, basal eGFR, admission eGFR, previous glycemia, CRP, LDH, lymphocytes, D-dimer, AST, and ALT are medians and IQR. Univariable and multivariable analyses are indicated as odds ratios, 95% confidence intervals, and *p*-values.

## Data Availability

The datasets generated during and/or analyzed during the current study are available from the corresponding author upon reasonable request.

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
