# Peer review of "Impact of Hypertension on COVID-19 Burden in Kidney Transplant Recipients: An Observational Cohort Study"

_viruses, 2022, doi:10.3390/v14112409_

Round 1

Reviewer 1 Report

In their manuscript, “Impact of hypertension on COVID-19 burden in kidney transplant recipients: an observational cohort study,” Aguiar-Brito, et al examine kidney transplant recipients from their institution who developed COVID-19 during the first part of the pandemic. While addressing the concomitant burden of hypertension and solid organ transplantation, there are several aspects of the manuscript that require attention. Specifically, the methods by which the cohort was developed, the comorbidities identified, and outcomes tracked is not well described. The development of the models is also unclear. Further, the outcomes models appear to all be stratified, however, the conclusions relate to a comparison of KTRs with and without hypertension.

1.       The population needs to be better defined, starting in the second paragraph of the study design and settings section. How were records reviewed? Was this an EHR based pull? We only tests at the home institution considered or were positive tests subsequently reported to the home hospital included, as well? How were all KTRs identified? Was there a timeframe from which the transplants had to occur?

2.       Why were only admitted patients examined? Most patients with COVID-19 are managed as outpatients, so admission to the hospital represents an important component in severity of illness assessment.

3.       Exclusion of individuals admitted to other hospitals represents a large potential source of bias. Was consideration given to at least comparing the baseline demographics and outcomes of the cohort admitted to other hospitals and those admitted to the home institution?

4.       How was race identified? Self-identification?

5.       How were comorbidities identified? ICD-9/10 codes? This is particularly important given hypertension is a key variable of interest in this study. What if a patient had hypertension prior to transplant that was resolved following transplantation? Was this person still considered hypertensive? Much more detail is required on how these variables were assessed.

6.       What are the models producing the ORs in Tables 1 and 2 representing? The statistical analysis section states that “variables that were univariately associated with a value of P<=0.1 were simultaneously included in a multivariate binary logistic regression model to estimate the ….” ORs between HTN and outcomes. These would seem to be for Table 3-5, so what are the models in Tables 1-2 adjusting for? Based on the ORs and 95% CIs, it appears that they are just univariate models providing little information beyond the P-values.

7.       In Table 3-4, what do the multivariable P-value relate to? They clearly represent an analysis of the variables with P<=0.1 in the univariate analysis, however, do align with the ORs and 95% CIs in the models on the right of the tables. This should be clarified.

8.       What consideration given to using ordinal logistic regression with outcomes tiered by severity to improve power?

9.       The authors report that only age remained significant in the model of need for HD among hypertensive KTRs, however, the 95% CI for donor type and liver disease do not cross 1. Again, clarification is needed in how these models were developed.

10.   The first line of the discussion states that hypertensive KTRs have higher rates of the evaluated outcomes of interest, however, I do not see a direct comparison of the rates between HTN and non-HTN KTR patients made. All models appear to be stratified. I’m curious if a parsimonious model with selective interaction of hypertensive status may be a more appropriate way to address this.

Minor

1.       In the abstract and introduction, the phrase “COVID-19 burden” is used, however, it appears that the authors are in fact referring to severity of illness.

2.       The syntax of the second sentence in the first paragraph of the introduction requires correction.

3.       First sentence of second paragraph of introduction, comorbidity should be plural.

4.       Third sentence of the second paragraph of the introduction does not make sense. SARS-CoV-2 is the causative agent of COVID-19 and therefore cannot also be an aggravating factor.

5.       Final paragraph of the introduction: how would these results inform disease cluster identification? The results inform individual patient care but not clustering of a communicable disease.

6.       The created models are multivariable, not multivariate.

Author Response

Dear reviewer, 

Please see attached our responses. 

Sincerely,

Érika B Rangel, MD, PhD

Reviewer 2 Report

  • Summary 

In the manuscript, Aguiar-Brito et al. performed a detailed retrospective analysis of 300 KTRs in a single center in Brazil. The authors verified the main risk factors related to hypertension (HTN) that contribute to COVID-19 progression and mortality in the KTR population. Specifically, they found that pre-existing HTN was independently associated with higher rates of mortality, transfer to the ICU, and AKI with hemodialysis (HD) requirement. In the hypertensive group, age, diabetes mellitus, heart disease, smoking, glycemic control before admission, C-reactive protein, lactate dehydrogenase, lymphocytes, and D-dimer were significantly associated with COVID-19 progression and mortality. Both lower basal and previous estimated glomerular filtration rates posed KTRs with HTN at greater risk for HD requirement.

The findings of this study are valuable in our understanding of the early identification factors that can be used to predict COVID-19 progression and mortality in KTRs affected by COVID-19, which is conducive to better patient flow management, therapeutic decisions, and allocation of resources.

  • General concept comments

The study is clear and relevant to the field. The study can be further improved by including the following suggestions listed in specific comments.

  • Specific comments:

1)      Page 2, the section of 2. Material and Methods, line 5 of the second paragraph, recommends adding “(DGF)” after “delayed graft function” since Figure 1. S. has the abbreviation of “DGF”.

2)      Suggest removing Figure 1. S., Table 1. S., Table 2. S., Table 3. S. from the main text to supplementary files.

3)       Page 3, Line 7, unnecessary punctuation here.

4)      Page 3, the section of 2.2. Demographic data, line 3, is smoking comorbidity?

5)      Page 4, the section of 2.4. Statistical analysis, line 5, spelling error(“do”), also, the full name of KDIGO should be given: Kidney Disease Improving Global Outcomes.

6)      Layout problem of “Table 1”. It is also suggested to include information of the clinical characteristics of these patients in Table 1 as well.

7)      Table 2, the study assessed the medical records of patients who were either hospitalized or non-hospitalized with the diagnosis of COVID19 during the study period. It would be better to include the information of outpatients/inpatients in Table 2 as well, since the status of outpatients/inpatients can also indicate COVID19 disease severity.

8)      Line 22-23, since the stage 3 AKI in HTN was also statistically different when compared to non-HTN individuals, it would also be helpful to investigate the main risk factors, including demographic data and laboratory parameters that were associated with this difference.

9)      Line 177, spelling error of the admission number here.

10)   Line 267, the full name of hs-troponin I should be given: High-Sensitivity Troponin I.

11)   Line 270, extra space between “particularly” and “antimetabolites”

Author Response

Dear Editor,

We would like to thank the reviewers for their comments. These comments were insightful and contributed to strengthening our manuscript. We provide a point-by-point answer to questions.   

Reviewer 2

  • Summary

In the manuscript, Aguiar-Brito et al. performed a detailed retrospective analysis of 300 KTRs in a single center in Brazil. The authors verified the main risk factors related to hypertension (HTN) that contribute to COVID-19 progression and mortality in the KTR population. Specifically, they found that pre-existing HTN was independently associated with higher rates of mortality, transfer to the ICU, and AKI with hemodialysis (HD) requirement. In the hypertensive group, age, diabetes mellitus, heart disease, smoking, glycemic control before admission, C-reactive protein, lactate dehydrogenase, lymphocytes, and D-dimer were significantly associated with COVID-19 progression and mortality. Both lower basal and previous estimated glomerular filtration rates posed KTRs with HTN at greater risk for HD requirement.

The findings of this study are valuable in our understanding of the early identification factors that can be used to predict COVID-19 progression and mortality in KTRs affected by COVID-19, which is conducive to better patient flow management, therapeutic decisions, and allocation of resources.

  • General concept comments

The study is clear and relevant to the field. The study can be further improved by including the following suggestions listed in specific comments.

  • Specific comments:

1)      Page 2, the section of 2. Material and Methods, line 5 of the second paragraph, recommends adding “(DGF)” after “delayed graft function” since Figure 1. S. has the abbreviation of “DGF”.

Response: We added the meaning of DGF, as pointed out.

2)      Suggest removing Figure 1. S., Table 1. S., Table 2. S., Table 3. S. from the main text to supplementary files.

Response: We added Figure 1S and Tables 1S-3S to the supplementary file.

3)       Page 3, Line 7, unnecessary punctuation here.

Response: We corrected the punctuation.

4)      Page 3, the section of 2.2. Demographic data, line 3, is smoking comorbidity?

Response: Thank you for pointing this out. Smoking is not comorbidity. We modified the sentence accordingly.

5)      Page 4, the section of 2.4. Statistical analysis, line 5, spelling error (“do”), also, the full name of KDIGO should be given: Kidney Disease Improving Global Outcomes.

Response: We added the full name of KDIGO and corrected the spelling error, as suggested.

6)      Layout problem of “Table 1”. It is also suggested to include information of the clinical characteristics of these patients in Table 1 as well.

Response: We reformulated the layout of Table 1 (we merged Table 1 and Table 2), as suggested.

7)      Table 2, the study assessed the medical records of patients who were either hospitalized or non-hospitalized with the diagnosis of COVID‐19 during the study period. It would be better to include the information of outpatients/inpatients in Table 2 as well, since the status of outpatients/inpatients can also indicate COVID‐19 disease severity.

Response: Thank you for your suggestion. Unfortunately, we did not have all the data from outpatients with COVID-19. The patients with mild COVID-19 were not hospitalized and were discharged from ER without collecting blood tests during the first wave. Others were monitored by phone calls from our telemedicine team. Comparison of outpatients/inpatients would be very informative in further studies not only for other waves of COVID-19 but also for other virus pandemics.       

8)      Line 22-23, since the stage 3 AKI in HTN was also statistically different when compared to non-HTN individuals, it would also be helpful to investigate the main risk factors, including demographic data and laboratory parameters that were associated with this difference.

Response: Thank you for your suggestion. Importantly, 109 out of 120 (91%) patients that developed AKI Stage 3 were submitted to hemodialysis. Therefore, the population of AKI stage 3 and hemodialysis represents essentially the same group of patients.  

9)      Line 177, spelling error of the admission number here.

Response: We corrected the spelling error.

10)   Line 267, the full name of hs-troponin I should be given: High-Sensitivity Troponin I.

 Response: The full name was added.

11)   Line 270, extra space between “particularly” and “antimetabolites”

Response: The extra space was deleted. 

Sincerely,

Érika B Rangel, MD, PhD